# Targeting *carA* Using Optimized Antisense Peptide Nucleic Acid–Cell-Penetrating Peptide Conjugates in *Acinetobacter baumannii*: A Novel Antibacterial Approach

**DOI:** 10.3390/ijms26199526

**Published:** 2025-09-29

**Authors:** Ju-Hui Seo, Yoo-Jeong Kim, Wook-Jong Jeon, Jung-Sik Yoo, Dong-Chan Moon

**Affiliations:** Division of Antimicrobial Resistance Research, National Institute of Infectious Diseases, National Institute of Health, Korea Disease Control and Prevention Agency, Cheongju-si 28159, Republic of Korea

**Keywords:** bactericidal, cell-penetrating peptides, recombinant protein, peptide nucleic acid, *Acinetobacter baumannii*

## Abstract

*Acinetobacter baumannii* is an opportunistic pathogen associated with severe bloodstream infections. It exhibits a high level of multidrug resistance, posing major clinical challenges. Antisense peptide nucleic acids (PNAs) represent a promising alternative to conventional antibiotics; however, their therapeutic efficacy depends on optimal delivery and molecular design. In this study, we aimed to enhance the antibacterial activity of PNAs against *A. baumannii* by systematically optimizing cell-penetrating peptides (CPPs), PNA length, target region, and chemical modifications. The efficacy and safety of CPP–PNA constructs were evaluated using a comprehensive set of approaches, including determination of minimum bactericidal and minimum inhibitory concentrations, quantitative reverse transcription polymerase chain reaction, Western blotting, and cytotoxicity assays. Three CPP–PNA constructs targeting *carA* were synthesized. Among these, the KFFK (FFK)_2_–PNA conjugate showed the strongest bacterial growth-inhibitory effect, while the addition of extra lysine residues reduced its efficacy. Further analyses showed that a 10-mer alpha (α)-PNA modification targeting the ribosomal binding site of *carA* had the greatest inhibitory effect. These results underscore the importance of rational CPP design and PNA optimization in developing effective antisense antimicrobials against *A. baumannii*.

## 1. Introduction

*Acinetobacter baumannii* is a Gram-negative opportunistic pathogen commonly associated with hospital-acquired infections and outbreaks. Owing to multi-drug resistance of *A. baumannii*, only a few antibiotics are effective against it [1]. Currently, colistin, which continues to be used despite its nephrotoxicity, remains one of the main therapeutic options; therefore, there is an urgent need for alternative approaches [2]. Protein delivery systems have diverse applications in biotechnology, including gene editing and drug delivery [3]. Several intracellular protein delivery systems have been developed for prokaryotic cells, with cell-penetrating peptides (CPPs) being among the most widely used. Various high-throughput screening methods have recently been explored to identify the most effective CPPs. Notably, a toehold switch-based “switch-on” reporter system has been introduced, enabling the real-time quantitative evaluation of multiple CPPs simultaneously [4]. To ensure that delivery systems are effective for in vivo applications, they must exhibit cell-type-specific targeting [5], low toxicity in mammalian cells [6], and efficient cellular uptake and cargo release [7]. To evaluate the efficiency of CPPs, it is necessary to compare multiple CPPs under the same experimental conditions [8].

CPPs constitute a widely recognized peptide-based delivery system that uses the KFFK (FFK)_2_ motif to enable the efficient translocation of impermeable molecules into prokaryotic cells [9]. Additionally, these peptides have been used to deliver molecular cargoes such as nanoparticles, proteins, peptides, and DNA into eukaryotic and prokaryotic cells, as their low molecular weight and high permeability make them effective drug delivery carriers [10]. While it is generally believed that cationic CPPs interact with the negatively charged components of bacterial membranes, creating transient pores to facilitate membrane passage, the precise mechanisms remain unclear [11]. Although the precise mechanisms of cellular uptake for CPP–peptide nucleic acid (PNA) conjugate drug delivery systems remain unclear, recent studies suggest that endocytosis and direct membrane penetration play important roles. Ghosal et al. identified SbmA as an inner-membrane transporter that facilitates the uptake of KFFK (FFK)_2_–PNA conjugates [12].

The disruption of carbamoyl-phosphate synthase small chain (*carA*) could compromise bacterial survival by affecting the pyrimidine biosynthesis pathway. Hence, *carA* was selected as the target for CPP–PNA conjugate binding [13]. However, the efficiency of CPP–PNA conjugates in targeting *A. baumannii* may be influenced by various factors, including the target gene site, PNA length, and chemical modifications [14]. Therefore, we aimed to optimize a specific approach to identify effective CPP–PNA conjugates against *A. baumannii*. In particular, our study demonstrates a comparative analysis of less extensively studied structures, such as α- and γ-modified PNAs.

## 2. Results

In this study, we tested whether CPP–PNA conjugates targeting the *carA* gene could inhibit the growth of *A. baumannii*. We first compared different CPP sequences and chemical modifications (α- and γ-PNA) to identify the most effective design. We then examined whether shortening the PNA length could improve activity. Finally, we confirmed the gene-silencing effects of the optimized conjugates at both the mRNA and protein levels, checked their selectivity against other bacterial strains, and evaluated their cytotoxicity.

### 2.1. Inhibition of Bacterial Cell Growth by CPP–PNA Conjugates

No *A. baumannii* colonies were detected in the group treated with 50 µM KFFK (FFK)_2_-PNA conjugate (Table 1). Similarly, KKFK (FFK)_2_ conjugated with PNA significantly inhibited *A. baumannii* growth, resulting in a colony count of 10.5 CFU at 50 µM (Table 1). The ribosomal binding site (RBS) region of *carA* was identified as the most effective target region. Among the chemical modifications tested, the alpha (α) modification showed better efficacy than the gamma modification at the same target site. Additionally, reducing the PNA length to 10-mer resulted in a two-fold improvement in minimum inhibitory concentrations (MICs). Importantly, the two control α-PNA modifications with slightly mismatched sequences showed antibacterial effects at the tested concentration of 12.5 µM. In comparison, CPP–PNA conjugates targeting other essential genes, including *acpP*, *ftsZ*, and *rne*, exhibited higher MICs (25–50 µM) than the α-PNA modification targeting *carA* (Table 2). Furthermore, the α-PNA modifications exhibited no inhibitory effect at concentrations exceeding 50 µM in specificity tests against standard Gram-positive and Gram-negative bacterial strains (Table 3). The inhibitory effect of α-PNA modifications targeting *carA* on *A. baumannii* was confirmed, showing a reduction in mRNA levels of approximately 70% at 25 µM (Figure 1). Notably, statistical significance of the inhibitory effects increased in a dose-dependent manner, except at concentrations of 3.13 and 12.5 µM. Western blot analysis further confirmed protein-level suppression, showing a seven-fold reduction in carA protein expression following treatment with 50 µM α-PNA compared with that in the untreated control (Figure 2).

### 2.2. Cytotoxicity of CPP–PNA Conjugates in Mammalian Cells

The cytotoxicity of α-PNA modifications at varying concentrations was assessed in human epithelial lung (A549) and laryngeal (HEp-2) cells to determine the safety profile of the peptides in mammalian systems. At 12.5 µM, the treatment group exhibited cytotoxic effects comparable to those of the positive control (10% DMSO), showing statistically significant toxicity in HEp-2 and A549 cells (*p* < 0.05; Figure 3).

## 3. Discussion

PNAs are non-polar molecules with a limited ability to independently cross bacterial membranes [12]. However, when conjugated to CPP, their intracellular delivery and effectiveness are enhanced, making them promising candidates for the development of antibacterial treatments [15]. For example, the peptide sequence KFFK (FFK)_2_ effectively facilitates PNA uptake via the inner membrane SbmA transporter protein [16]. Moreover, the KFFK (FFK)_2_ peptide improves membrane permeability, allowing PNA to enter the cytoplasm and exert its targeted antibacterial effects [16].

The effectiveness of CPP–PNA conjugate delivery into bacterial cells varies significantly depending on the bacterial species and strain and is influenced by differences in surface properties such as membrane composition and charge [17]. Cationic CPPs generally perform better than neutral or anionic ones, as their positive charge enables them to interact more effectively with the negatively charged components of bacterial membranes [18,19]. The increased efficiency of positively charged CPPs is attributable to their ability to readily bind to negatively charged phospholipids and lipopolysaccharides present in bacterial cell membranes [17,19].

Here, positively charged CPPs, KKFK (FFK)_2_ and KKKK (FFK)_2_, were designed to increase their penetration efficiency into *A. baumannii*. However, an increase in the number of positively charged residues did not increase membrane permeability. For example, KKKK (FFK)_2_, despite having the highest number of lysine residues, showed no bactericidal activity in minimum bactericidal concentration (MBC) assays. CPP efficiency is influenced by multiple factors, including disorder propensity, net charge, isoelectric point, and hydrophobicity [17,19], rather than solely by the number of cationic residues. This multifactorial influence may explain the results observed in the present study.

Here, KFFK (FFK)_2_-conjugated PNA targeting *carA* showed effective cellular entry and antisense activities. Rose et al. obtained a similar result using a CPP (RXR)_4_XB-conjugated PNA oligomer, which also targeted *carA* and resulted in *A. baumannii* growth inhibition [13].

Notably, the combination of KFFK (FFK)_2_ with *carA*-targeted PNA was the most potent inhibitor of bacterial growth, as evidenced by our MBC results.

MICs varied depending on PNA design, even with the same CPP. Among the target regions, PNA bound to RBS exhibited the highest efficacy. Previous studies have reported that targeting the start codon and RBS led to the most effective inhibition, with increased MIC values observed when downstream regions were targeted [14]. Here, *carA*-targeted PNA incorporating four nucleotides of the start codon and RBS demonstrated the highest efficacy. However, in contrast to prior study findings, targeting the downstream stop codon resulted in a lower MIC than targeting the start codon, indicating that binding site selection does not always directly correlate with inhibitory effectiveness. This observation is consistent with the findings of a previous study [20], showing that even among PNAs targeting the same start codon region, a shift in just two nucleotides in the binding position can lead to substantial differences in MICs. These results highlight the importance of optimizing the binding position while designing CPP–PNA conjugates.

Gamma (γ)-PNAs, a backbone-modified PNA, exhibit significantly higher cellular uptake than unmodified PNAs, and this is likely due to their helical conformation that facilitates cellular delivery [21]. Additionally, γ-PNAs demonstrate high binding affinity and precise targeting of genomic DNA as well as coding and non-coding RNAs, making them a strong candidate for antisense therapeutic drug development. The enhanced physicochemical properties, including increased solubility, further support the potential of γ-PNAs in diverse biomedical applications [22].

In this study, although both α and γ modifications involved the addition of cationic amino acids, γ modifications had no significant effect on the MIC, whereas α modifications at two positions within the PNA sequence led to a two-fold reduction in the MIC. Notably, α-PNA modifications also enhanced the stability of PNA duplexes with DNA and RNA while maintaining sequence specificity, thereby improving binding affinity and antibacterial activity [23]. The superior bactericidal activity of α-PNA over γ-PNA in *carA*-targeted PNA suggests that α modification is a more effective structural optimization strategy.

Previous studies have explored the relationship between PNA length and MIC values in Gram-negative bacteria, identifying 12-, 11-, and 10-mer PNAs as the most used [14,24]. Among these, 10-mer PNA was found to be the most effective. In another study, 11-mer PNA demonstrated the highest activity (average MIC: 3.6 μM), followed by 10-mer (4.1 μM) and 12-mer (5.6 μM) PNAs [14]. Consistent with these findings, our study showed that reducing PNA length from 13 to 10 nucleotides led to a two-fold decrease in the MIC. Furthermore, the mRNA-silencing effect of the 10-mer α-PNA targeting *carA* was confirmed using quantitative real-time polymerase chain reaction (qRT-PCR) analysis, which showed a significant reduction in *carA* transcript levels. The 10-mer α-PNA is known to bind to the RBS, interfering with ribosome assembly and consequently exposing the mRNA to degradation pathways such as nonsense-mediated decay [20]. Western blot analysis further validated the inhibitory effect at the protein level, showing significant suppression of carA protein expression following treatment with 50 µM 10-mer α-PNA. Additionally, mismatched versions of the 10-mer α-PNA displayed a higher MIC of 12.5 µM, and their activity was diminished in other bacterial species, including *Escherichia coli*, *Pseudomonas aeruginosa*, and *Staphylococcus aureus* (MIC ≥ 50 µM), indicating high specificity toward *A. baumannii*.

Although the 13-mer PNA could bind to only one site in the *carA* gene, the 10-mer PNA could target 37 different sites (Appendix A). This broader binding range likely increased the probability of interfering with additional essential genes, such as *parC*, thereby contributing to the observed reduction in the MIC. The residual antibacterial activity of the mismatched 10-mer α-PNA is also likely due to its partial hybridization with other target genes (Appendix A). However, this broad-spectrum targeting raises concerns about potential off-target effects, including unintended interactions with the normal microbiota or host cells, and thus requires further investigation.

Here, CAMHB was used as the growth medium, and *A. baumannii* ATCC 17978 was employed as the test strain, which may account for the differences in findings between this study and a previous study [20]. Nevertheless, the 10-mer α-PNA modifications targeting *carA* demonstrated 4- to 8-fold greater efficacy compared to previously reported CPP–PNA conjugates targeting *acpP*, *ftsZ*, and *rne*, all of which are known antimicrobial targets [20].

In the cytotoxicity assay, an 84% reduction in HEp-2 cell viability was observed at a concentration of 12.5 µM, demonstrating higher cytotoxicity than that in a previous study [20]. This finding indicates that toxicity occurred at a much lower concentration than for conventional antibiotics, which typically show cytotoxic effects at concentrations between 150 and 300 mg/L [25]. Furthermore, at 25 µM, α-PNA modifications exhibited strong cytotoxic effects in both HEp-2 and A549 cells, comparable to those of the negative control.

The presence of cationic residues, such as lysine and arginine, enhances CPP membrane penetration but also increases cytotoxicity due to membrane disruption [19]. In contrast, other studies have reported that short CPPs, such as 8-mer sequences, exhibit minimal cytotoxicity and immunogenicity [26]. Previous research has also shown that cytotoxicity varies depending on the length of the CPP [27], suggesting that further optimization of CPP sequence and length, or the development of novel bacterial membrane-penetrating agents, is necessary to improve safety profiles.

Here, the 13-mer PNA targeting the RBS region of *carA* showed an MIC of 25 µM. In contrast, the α-modified 10-mer PNA demonstrated effective antibacterial activity even at 6.25 µM, emphasizing the importance of structural modification and sequence optimization in reducing MIC values for therapeutic applications. Although further studies are required to evaluate the efficacy against clinical isolates in murine infection models of the α-modified 10-mer PNA, our findings suggest that optimization of CPP–PNA conjugates may serve as a key strategy for improving efficacy. In this study, we used covalently linked CPP–PNA conjugates to target the *carA* gene in *A. baumannii*. Previous work by Morris et al. showed that non-covalent peptide carriers, such as Pep-3, can effectively deliver DNA mimics and achieve target-specific gene suppression without covalent linkage. This suggests that both covalent and non-covalent peptide strategies are viable for PNA delivery. Furthermore, amphipathic design and stabilization strategies, like PEGylation, could enhance CPP–PNA stability and uptake, offering a potential approach to improve efficacy against *A. baumannii* [28].

Despite these promising results, the application of PNAs is limited by challenges such as complex synthesis and high production costs. However, large-scale synthesis reduces cost by more than half, and previous studies have proposed combination therapy with synergistic antibiotics such as colistin as a feasible strategy [29]. Nevertheless, optimizing CPP–PNA conjugates remains a crucial step in developing effective alternatives to conventional antibiotics.

Our results indicate that CPP–PNA conjugates with enhanced antimicrobial activity, specifically optimized for *A. baumannii*, can be developed.

## 4. Materials and Methods

### 4.1. MBC of CPP–PNA Conjugates for A. baumannii

Artificially synthesized CPP–PNA (*carA*, TCAAACCAAAGCT) conjugates were used to silence *carA*. Briefly, *carA*-specific oligonucleotides were identified from the genomic sequence of *A. baumannii* ATCC 17978 (CP049363.1). The CPP–PNA conjugates, namely KFFK (FFK)_2_–PNA, KKFK (FFK)_2_–PNA, and KKKK (FFK)_2_–PNA, were synthesized by HLB PANAGENE Co., LTD. (Daejeon, Republic of Korea). These conjugates were engineered to specifically bind to the translation initiation region of *carA*, which includes the TTG start codon and the Shine-Dalgarno sequence (TTTGGT) crucial to ribosomal interaction.

To prepare bacterial suspensions, cultures of *A. baumannii* ATCC 17978 were grown until the OD_600_ value reached 0.8. The CPP–PNA conjugates were prepared to achieve concentrations of 50 and 0 µM in a total volume of 100 µL per well in 96-well plates. Subsequently, 10 µL of the adjusted bacterial suspension was inoculated into each well (1 × 10^7^ CFU/mL). The plates were then incubated at 37 °C for 4 h. Subsequently, 50 µL of each CPP–PNA conjugate-treated bacterial suspension was spread on Muller–Hinton agar plates, with each condition distributed across two plates. The plates were subsequently incubated at 37 °C for 24 h to enable bacterial growth and colony formation.

### 4.2. MIC of Different CPP–PNA Conjugates for A. baumannii

CPP–PNA conjugates were synthesized by HLB PANAGENE Co., LTD. The binding sites within *A. baumannii* ATCC 17978 were predicted and analyzed using Geneious Prime (version 1 February 2025), enabling precise identification of target regions for subsequent antisense PNA design. Specifically, α-PNA modifications introduced arginine, while γ-PNA modifications incorporated guanidino-lysine. The diverse CPP–PNA and α- and γ-modified PNA oligomers were generated via solid-phase peptide synthesis using benzothiazole sulfonyl and standard fluorenylmethyloxycarbonyl chemistry procedures on an automatic synthesizer (HLB-PANAGENE). The synthesized PNA oligomers were cleaved from the solid support resin using a mixed solution containing trifluoroacetic acid. The PNA mixture separated from the solid support was purified using reverse-phase high-performance liquid chromatography, and the purity and molecular weight were determined and confirmed using high-performance liquid chromatography and matrix-assisted laser desorption/ionization time-of-flight mass spectrometry (Appendix A). The MIC of each CPP–PNA conjugate was determined using the broth microdilution method in accordance with the Clinical and Laboratory Standards Institute guidelines. *A. baumannii* ATCC 17978 was used as the test strain. The CPP–PNA conjugates were tested at concentrations of 0, 1.625, 3.125, 6.25, 12.5, 25, and 50 µM in 100 µL of cation-adjusted Mueller–Hinton broth with N-Tris (hydroxymethyl) methyl-2-aminoethanesulfonic acid buffer (CAMHB; T3462; Thermo Fisher Scientific, Waltham, MA, USA). The microdilution plates were incubated at 37 °C for 18–20 h. A minimum of two independent experiments, each performed in duplicate, were conducted to ensure reproducibility.

### 4.3. qRT-PCR

*A. baumannii* 17978 was cultured in Muller–Hinton broth (MHB) at 37 °C to an optical density of approximately 1.0 × 10^8^ CFU/mL. The α-PNA modification, namely KFFK (FFK)_2_-AAC^CAAA^GCT, was administered at final concentrations of 50, 25, 12.5, 6.25, 3.13, and 1.56 µM, respectively, with untreated cultures serving as negative controls. After a 4 h incubation period, total RNA was extracted using an RNeasy Mini Kit (Qiagen, Hilden, Germany), per the manufacturer’s protocol, followed by treatment with RNase-Free DNase Set (Qiagen) to remove genomic DNA contamination. First-strand cDNA was synthesized using RNA to cDNA EcoDry Premix (Takara, San Jose, CA, USA) with random hexamers. qRT-PCR was performed on a 7500 Fast Real-Time PCR System (Applied Biosystems, Foster City, MA, USA) using THUNDERBIRD™ Next SYBR^®^ qPCR Mix (TOYOBO, Osaka, Japan). Each 20 µL reaction mixture contained 10 µL of 2× master mix, 0.4 µM of each primer, and 2 µL of cDNA. Thermal cycling conditions included initial denaturation at 95 °C for 20 s, followed by 40 cycles at 95 °C for 3 s and 60 °C for 30 s. Gene expression was normalized to the *rpoB* level, and relative expression levels were calculated using the comparative delta-delta Ct (2^−ΔΔCt^) method. Primer sequences are listed in Table 4.

### 4.4. Western Blotting

*A. baumannii* ATCC 17978 was cultured in MHB at 37 °C to approximately 1.0 × 10^8^ CFU/mL. The α-PNA modification, namely KFFK (FFK)_2_-AAC^CAAA^GCT, was administered at final concentrations of 50 µM with untreated cultures serving as negative controls during 4 h. Purified recombinant carA protein was custom produced by Engitein (Seoul, Korea). *carA* was amplified and cloned into pET-28a (+) for expression of an N-terminal His-tagged recombinant protein. The plasmid was transformed into *E. coli* BL21 (DE3), and protein expression was induced with 0.5 mM IPTG at 37 °C for 4 h. The cells were harvested, lysed using sonication, and soluble fractions purified using Ni-NTA affinity chromatography (Qiagen) under native conditions. Protein purity and size were determined using SDS–PAGE. Polyclonal antibodies against the carA protein were custom-produced by YNTOAB (Gyeonggi, Republic of Korea) via subcutaneous immunization of New Zealand White rabbits with purified recombinant protein emulsified in Freund’s adjuvant. For initial immunization, 0.5 mg of recombinant carA protein was emulsified in complete Freund’s adjuvant and administered to the rabbits via subcutaneous injection. Subsequent booster injections contained 0.2 mg of protein emulsified in incomplete Freund’s adjuvant and were administered biweekly for a total of three times. Blood was collected from the marginal ear vein, and sera were separated by centrifugation. Antibody titers were evaluated using indirect ELISA against the purified recombinant protein to confirm successful immunization. Sera were collected 1 week after the final boost. For Western blotting, whole-cell lysates and recombinant protein were separated using SDS–PAGE, transferred onto PVDF membranes (Bio-Rad, Hercules, CA, USA), blocked, and incubated with anti-carA polyclonal or anti-OmpA monoclonal antibodies diluted 1:1000. Horseradish peroxidase-conjugated secondary antibodies (Bio-Rad) diluted 1:2000 and enhanced chemiluminescence reagents (Intron, Seongnam-si, Republic of Korea) were used for detection. OmpA was used as a loading control.

### 4.5. Evaluation of CPP–PNA Conjugate Cytotoxicity

HEp-2 and A549 cells, derived from human laryngeal and lung epithelium, respectively, were used as mammalian model systems. HEp-2 cells were maintained in Dulbecco’s modified Eagle medium, whereas A549 cells were grown in Roswell Park Memorial Institute 1640 medium (GenDEPOT, Altair, TX, USA) supplemented with 10% fetal bovine serum and 1% penicillin–streptomycin (Gibco, Grand Island, NY, USA).

For the cytotoxicity assay, HEp-2 (1 × 10^4^ cells/100 µL) and A549 (5 × 10^3^ cells/100 µL) cells were seeded in 96-well plates and incubated at 37 °C in a 5% CO_2_ atmosphere. After 24 h, the cells were washed with 1× PBS and exposed to 50, 25, 12.5, 6.25, 3.13, and 1.56 μM of α-PNA modification, namely KFFK (FFK)_2_-AAC^CAAA^GCT, for an additional 24 h. DMSO (10%) and untreated cells served as positive and negative controls, respectively. Cell viability was assessed using the CCK-8 assay kit (SIGMA, St. Louis, MO, USA), following the manufacturer’s instructions. The fluorescence signal of CCK-8 was measured at 450 nm using a microplate reader (TECAN Infinite M200; Männedorf, Switzerland). All assays were performed in biological duplicates, and data are reported as the mean ± standard error.

### 4.6. Statistics and Reproducibility

CPP efficiency and CPP–PNA conjugate cytotoxicity were analyzed using two-way analysis of variance followed by Dunnett’s multiple comparisons test, performed with the IBM SPSS Statistics software version 29.0 (IBM, New York, NY, USA). The *carA* mRNA expression levels were analyzed using a one-way analysis of variance followed by Dunnett’s multiple comparisons test using IBM SPSS Statistics. The results are expressed as the mean ± standard error of the mean. Statistical significance was assessed using a paired *t*-test with significance thresholds set at *p* < 0.05.

## 5. Conclusions

In this study, optimized KFFK(FFK)_2_ CPP–PNA conjugates targeting the *carA* gene showed strong antibacterial activity against *A. baumannii*. Their efficacy was strongly influenced by the choice of target site, chemical modification, and PNA length. Notably, the 10-mer PNA was more effective than the 13-mer design, likely because it can suppress not only *carA* but also other essential bacterial genes. We also found that α-modified PNAs targeting the ribosome-binding site (RBS) of *carA* were more potent than γ-modified PNAs. However, cytotoxicity was observed at 12.5 µM, which appeared to be mainly due to the CPP moiety. To reduce this toxicity, future studies should explore CPP length optimization or the development of new membrane-penetrating carriers. Overall, our findings support CPP–PNA conjugates as a promising strategy against *A. baumannii* and provide useful guidance for the design of next-generation antisense antibacterial agents.

## Figures and Tables

**Figure 1 ijms-26-09526-f001:**
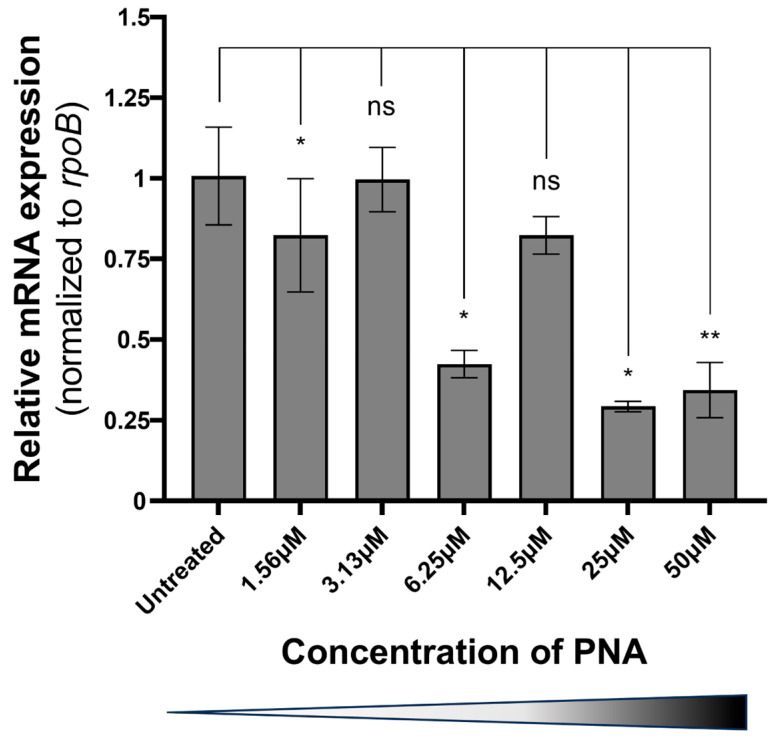
The relative expression level of the *carA* transcript in *A. baumannii* ATCC 17978 was analyzed using quantitative real-time PCR across different peptide nucleic acid (PNA) treatment concentrations. Asterisks (*) indicate statistically significant values (*p* < 0.05), with an increasing number of asterisks denoting greater statistical significance. “ns” means not significant.

**Figure 2 ijms-26-09526-f002:**
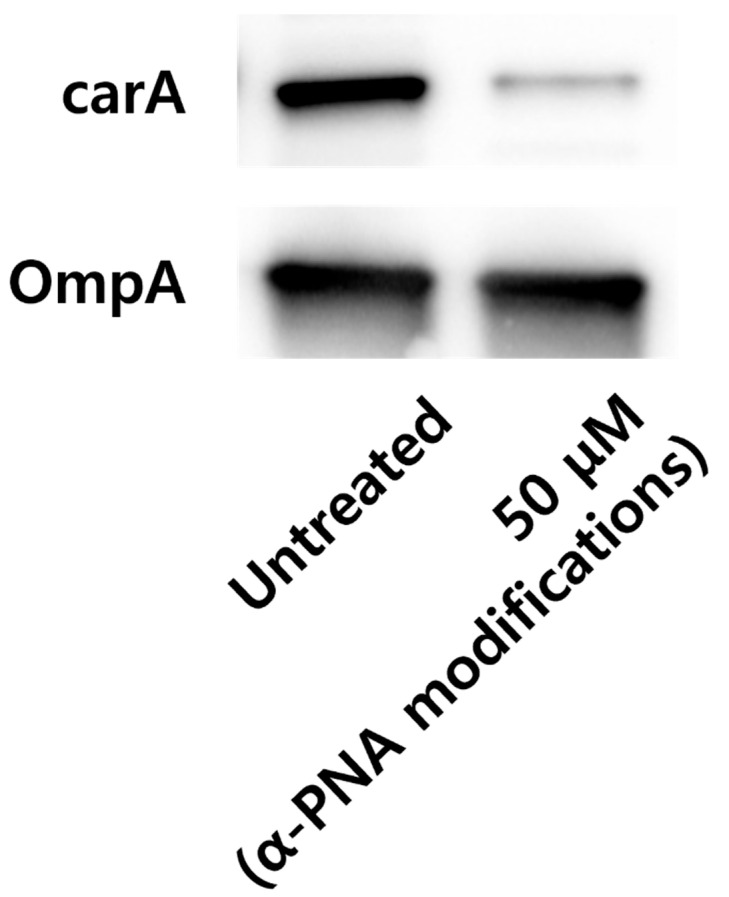
Western blot image of carA protein levels in *A. baumannii* treated with α-PNA modifications targeting the *carA* gene.

**Figure 3 ijms-26-09526-f003:**
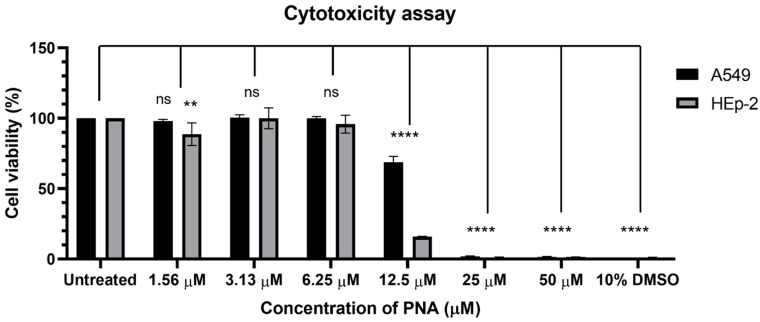
Cytotoxicity of the CPP-PNA conjugate in human cell lines. Cytotoxicity of α-PNA modifications in A549 and HEp-2 cells. DMSO (10%) was used as a positive control, and untreated cells served as negative controls. Statistical significance was determined by comparing the cytotoxicity of A549 and HEp-2 cells at the same concentration with that of the negative control. Asterisks (*) indicate statistically significant values (*p* < 0.05), with an increasing number of asterisks denoting greater statistical significance. “ns” means not significant.

**Table 1 ijms-26-09526-t001:** Minimum bactericidal concentrations of CPP–PNA.

CPP Sequence	PNA Sequence	MBC (CFU) in 50 µM CPP–PNA	Average CFU
KFFK(FFK)_2_	TCAAACCAAAGCT	0	0
	TCAAACCAAAGCT	0	
KKFK(FFK)_2_	TCAAACCAAAGCT	7	10.5
	TCAAACCAAAGCT	14	
KKKK(FFK)_2_	TCAAACCAAAGCT	TNTC	TNTC
	TCAAACCAAAGCT	TNTC	

TNTC, too numerous to count; MBC, Minimum bactericidal concentration; CPP–PNA, cell-penetrating peptide–peptide nucleic acid; CFU, colony-forming unit.

**Table 2 ijms-26-09526-t002:** MICs of diverse CPP–PNAs against *A. baumannii*.

Gene Target Region	CPP	PNA Sequence	PNA Length (mer)	MIC (μM)	Modification
*carA* (RBS)	KFFK(FFK)_2_	TCAAACCAAAGCT	13	25	-
*carA* (Start codon)	KFFK(FFK)_2_	CGGGGGTGCTCAA	13	>50	-
*carA* (Stop codon)	KFFK(FFK)_2_	TTACTGTTTAGAT	13	50	-
*carA* (RBS)	KFFK(FFK)_2_	TCAAA*CCA*AAGCT	13	25	γ
*carA* (RBS)	KFFK(FFK)_2_	TCAAA^CCA^AAGCT	13	12.5	α
*carA* (RBS)	KFFK(FFK)_2_	AAC^CAAA^GCT	10	6.25	α
Control 1 (mismatch)	KFFK(FFK)_2_	ACC^CACA^GCT	10	12.5	α
Control 2 (mismatch)	KFFK(FFK)_2_	ACC^CACA^CCC	10	12.5	α
Control 3 (*acpP*)	KFFK(FFK)_2_	TGATTTGCCAC	10	25	
Control 4 (*ftsZ*)	KFFK(FFK)_2_	GAGGCCATGAC	10	50	
Control 5 (*rne*)	KFFK(FFK)_2_	ACGTTTCATGG	10	50	

*, Gamma (γ) modification sites; ^, Alpha (α) modification sites: MIC, Minimum inhibitory concentration; CPP–PNA, cell-penetrating peptide–peptide nucleic acid.

**Table 3 ijms-26-09526-t003:** MIC of CPP–PNA against various isolates of Gram-negative and Gram-positive bacterial species.

Gene Target Region	CPP	PNA Sequence	MIC (μM)		
			*Escherichia coli* ATCC 25922	*Staphylococcus aureus* ATCC 29213	*Pseudomonas aerusinosa* ATCC 27853
*carA* (RBS)	KFFK(FFK)_2_	AAC^CAAA^GCT	>50	>50	50

^, Alpha (α) modification sites; MIC, Minimum inhibitory concentration; CPP–PNA, cell-penetrating peptide–peptide nucleic acid.

**Table 4 ijms-26-09526-t004:** qRT-PCR primer sequences used in this study.

Primer Name	Oligonucleotide Sequence (5′ to 3′)	Reference
CPP–PNA-F (*carA*)	GGTCTGAAGGTTCATGGGTATT	This study
CPP–PNA-R (*carA*)	GCAGGGACAACAGTGAGTTTA	This study
rpoB-F	TGACTCTGGTGTGTGTGTAATC	This study
rpoB-R	CACCTGCAACCATTTCATCTTC	This study

CPP–PNA, cell-penetrating peptide–peptide nucleic acid; qRT-PCR, quantitative reverse transcription polymerase chain reaction.

## Data Availability

All datasets generated and analyzed in this study are included in the manuscript and the Appendix A.

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
