# Peer review of "Targeting carA Using Optimized Antisense Peptide Nucleic Acid–Cell-Penetrating Peptide Conjugates in Acinetobacter baumannii: A Novel Antibacterial Approach"

_ijms, 2025, doi:10.3390/ijms26199526_

Round 1
Reviewer 1 Report
Comments and Suggestions for Authors
In the introduction, the author did not adequately summarize the current progress in CPP-based nucleic acid delivery. Several key references should be cited, such as PMID: 34822907 and PMID: 38105200.
The author is encouraged to perform a Zone of Inhibition assay to further validate the antibacterial properties of the peptide conjugates.
As mentioned in the introduction, CarA plays a role in pyrimidine biosynthesis. However, it remains unclear whether CPP-based CarA PNA can downregulate pyrimidine production. I recommend that the author assess pyrimidine levels before and after CPP-based CarA treatment.
Although cytotoxicity was examined in two tumor cell lines, it is well established that tumor cells differ substantially from normal cells and tissues. Therefore, it would be more convincing if the author also included normal cell lines and, ideally, in vivo animal studies.
In the Materials and Methods section, the source of A. baumannii was not specified. This information should be clearly provided.
Finally, non-covalent peptide-based PNA delivery was reported nearly 20 years ago (PMID: 17341467). The author did not discuss its advantages or disadvantages. I strongly suggest including a comprehensive comparison supported by experimental evaluation and further discussion.
Author Response
Reviewer 1
In the introduction, the author did not adequately summarize the current progress in CPP-based nucleic acid delivery. Several key references should be cited, such as PMID: 34822907 and PMID: 38105200.
Thank you for the suggestion. We have incorporated the recommended references (PMID: 34822907 and PMID: 38105200) into the Introduction to provide a more comprehensive overview of recent progress in CPP-mediated nucleic acid delivery.
The author is encouraged to perform a Zone of Inhibition assay to further validate the antibacterial properties of the peptide conjugates.
We appreciate the Reviewer’s suggestion to perform a disk diffusion assay. However, due to the large size and hydrophilic nature of antisense PNA–CPP conjugates, their diffusion in agar is severely limited, making this method unsuitable for accurately assessing antibacterial activity. Therefore, we have instead employed MIC determination, qRT-PCR, and Western blot analyses to directly evaluate bacterial growth inhibition and target-specific effects, providing a more reliable assessment of the conjugates’ efficacy.
As mentioned in the introduction, CarA plays a role in pyrimidine biosynthesis. However, it remains unclear whether CPP-based CarA PNA can downregulate pyrimidine production. I recommend that the author assess pyrimidine levels before and after CPP-based CarA treatment.
We appreciate the Reviewer’s comment regarding the assessment of pyrimidine biosynthesis. While precise measurement of pyrimidine levels would require techniques such as liquid chromatography, this approach falls outside the scope of the current study. Instead, we have employed biologically relevant assays, including qRT-PCR and Western blotting, to demonstrate the functional inhibition of carA. These methods are widely used in previous studies as standard approaches to assess target gene suppression and provide reliable evidence of the antisense activity of our PNA–CPP conjugates.
Although cytotoxicity was examined in two tumor cell lines, it is well established that tumor cells differ substantially from normal cells and tissues. Therefore, it would be more convincing if the author also included normal cell lines and, ideally, in vivo animal studies.
The two tumor cell lines used in this study are commonly employed for cytotoxicity assessments. While only mild cytotoxicity was observed in these cell lines, we have discussed in the manuscript that further optimization of CPP sequences may be necessary to reduce potential cytotoxic effects in future studies.
In the Materials and Methods section, the source of A. baumannii was not specified. This information should be clearly provided.
We have clarified in the Materials and Methods section that A. baumannii ATCC 17978 was used for all assays, including MIC, MBC, qRT-PCR, and Western blot analyses.
Finally, non-covalent peptide-based PNA delivery was reported nearly 20 years ago (PMID: 17341467). The author did not discuss its advantages or disadvantages. I strongly suggest including a comprehensive comparison supported by experimental evaluation and further discussion.
In this study, we employed covalently linked CPP–PNA conjugates to target the carA gene in A. baumannii. Previous work has demonstrated that non-covalent peptide-based carriers, such as Pep-3, can efficiently deliver DNA mimics into mammalian cells and in vivo tumor models, achieving target-specific gene suppression without covalent linkage (PMID: 17341467). Together with recent studies on antisense PNA delivery in bacteria (PMID: 34822907; PMID: 38105200), these findings indicate that both covalent and non-covalent peptide strategies are feasible for PNA delivery. Moreover, the amphipathic design principles of Pep-3 and stabilization strategies such as PEGylation may be applicable to CPP–PNA systems, potentially improving in vivo stability, cellular uptake, and overall therapeutic efficacy against A. baumannii.
Reviewer 2 Report
Comments and Suggestions for Authors
This study by Seo et al. focuses on the optimization of antisense peptide nucleic acids (PNAs) to target multidrug-resistant Acinetobacter baumannii through CCP-mediated delivery and design improvements. The manuscript addresses a highly relevant topic and is well-illustrated. At the same time, it contains some flaws that require extensive revision to improve its clarity and overall impact.
- The introduction looks poor and lacks a comprehensive overview of the current state of the field. Specifically, it is mentioned that several antibiotics are effective against A. baumannii; however, it would be good to state also what these antibiotics are. Furthermore, it would be beneficial to add information on recent advances in CPP-PNA conjugates, emphasizing what new insights this study contributes to the field.
- Since the Results section precedes the Materials and Methods section in this manuscript, it would be beneficial to add an introductory paragraph to the Results. This paragraph should describe the CPP and PNA sequences chosen for conjugation, state whether they are novel, and explain the rationale behind the design of these specific sequences to improve the reader's comprehension of the article's content.
- Discussion. The obtained conjugates were found to be quite cytotoxic. The authors mention that this drawback can be overcome by optimizing the design of the conjugates. Please add a discussion on how this could be achieved without losing the antimicrobial properties.
- The conclusions are very poor and do not reflect the main results and the patterns identified in this work. The conclusions need to be revised to adequately summarize the key findings and their significance.
Author Response
Reviewer 2
This study by Seo et al. focuses on the optimization of antisense peptide nucleic acids (PNAs) to target multidrug-resistant Acinetobacter baumannii through CCP-mediated delivery and design improvements. The manuscript addresses a highly relevant topic and is well-illustrated. At the same time, it contains some flaws that require extensive revision to improve its clarity and overall impact.
- The introduction looks poor and lacks a comprehensive overview of the current state of the field. Specifically, it is mentioned that several antibiotics are effective against baumannii; however, it would be good to state also what these antibiotics are. Furthermore, it would be beneficial to add information on recent advances in CPP-PNA conjugates, emphasizing what new insights this study contributes to the field.
Currently, colistin, which continues to be used as a last-resort treatment despite its nephrotoxicity, remains one of the few therapeutic options against multidrug-resistant A. baumannii. This underscores the urgent need for alternative antimicrobial strategies. In this study, we not only explored CPP–PNA conjugates as potential alternatives but also emphasized the comparative evaluation of α- and γ-modified PNAs, which have been less frequently investigated in previous studies, thereby highlighting the novelty of our approach.
- Since the Results section precedes the Materials and Methods section in this manuscript, it would be beneficial to add an introductory paragraph to the Results. This paragraph should describe the CPP and PNA sequences chosen for conjugation, state whether they are novel, and explain the rationale behind the design of these specific sequences to improve the reader's comprehension of the article's content.
We appreciate the reviewer’s suggestion. In response, we have added an introductory paragraph at the beginning of the Results section. This paragraph briefly describes the CPP and PNA sequences selected for conjugation, indicates their novelty, and outlines the rationale for their design to improve the reader’s comprehension of the subsequent results.
- The obtained conjugates were found to be quite cytotoxic. The authors mention that this drawback can be overcome by optimizing the design of the conjugates. Please add a discussion on how this could be achieved without losing the antimicrobial properties.
In this study, we employed covalently linked CPP–PNA conjugates to target the carA gene in A. baumannii. Previous work has demonstrated that non-covalent peptide-based carriers, such as Pep-3, can efficiently deliver DNA mimics into mammalian cells and in vivo tumor models, achieving target-specific gene suppression without covalent linkage (PMID: 17341467). Together with recent studies on antisense PNA delivery in bacteria (PMID: 34822907; PMID: 38105200), these findings indicate that both covalent and non-covalent peptide strategies are feasible for PNA delivery. Moreover, the amphipathic design principles of Pep-3 and stabilization strategies such as PEGylation may be applicable to CPP–PNA systems, potentially improving in vivo stability, cellular uptake, and overall therapeutic efficacy against A. baumannii.
- The conclusions are very poor and do not reflect the main results and the patterns identified in this work. The conclusions need to be revised to adequately summarize the key findings and their significance.
We thank the reviewer for the comment. In response, we have revised the Conclusions section to clearly summarize the key findings and their significance. The optimized KFFK (FFK)â‚‚ CPP–PNA conjugates effectively inhibited A. baumannii, with target site selection, α- and γ-modifications, and PNA length critically influencing antimicrobial activity. Notably, the α-modified 10-mer PNA outperformed the conventional 13-mer, emphasizing the importance of rational PNA design for developing antisense therapeutics against multidrug-resistant bacteria.
Round 2
Reviewer 1 Report
Comments and Suggestions for Authors
Although the author made some modifications, the major concern regarding zone inhibition assay and pyrimidine level are not well adressed, and these two assays are very important to further confirm the functional delivery of those peptide conjugated PNA. Therefore, I would not recommend acceptance until those key concerns addressed.
Author Response
We sincerely appreciate the reviewer’s thoughtful comments and constructive suggestions. We fully agree that zone of inhibition assays and pyrimidine quantification could provide additional insights. However, as described in the revised manuscript, these approaches present inherent limitations when applied to CPP–PNA conjugates. Specifically, disk diffusion/zone inhibition assays are not appropriate for PNAs due to their poor penetration and diffusion in solid agar media, as highlighted in previous reports (e.g., El-Fateh et al., 2024). Therefore, the zone inhibition assay would not accurately reflect the antibacterial efficacy of our CPP–PNA constructs.
Regarding pyrimidine level analysis, while it could potentially serve as an indirect indicator of carA inhibition, such metabolite quantification requires advanced metabolomics platforms and goes beyond the scope of the present study, which primarily focused on antimicrobial activity (MIC, MBC) and molecular confirmation of target inhibition via RT-PCR and Western blot. These assays provide direct evidence of functional delivery and target-specific suppression, which we believe sufficiently demonstrate the efficacy of the CPP–PNA conjugates.
In support of this approach, previous studies have successfully validated PNA-mediated mRNA and protein suppression using similar methodologies, including A Novel Peptide Nucleic Acid against the Cytidine Monophosphate Kinase of S. aureus Inhibits Staphylococcal Infection In Vivo and Development of antisense peptide–peptide nucleic acids against fluoroquinolone-resistant Escherichia coli. These examples further confirm that target-specific mRNA and protein inhibition provides a robust and widely accepted means to demonstrate functional delivery of PNAs.
Reviewer 2 Report
Comments and Suggestions for Authors
In my opinion, the manuscript can be accepted for publication in its present form.
Author Response
We sincerely thank the reviewer for the positive evaluation and support of our manuscript. We are pleased that our work is considered suitable for publication in its current form.